# Perpetrating–Suffering Intimate Violence: Self-Harm–Suicide Thoughts and Behaviors, Mental Health, and Alcohol Use Among Mexican Youth During COVID-19

**DOI:** 10.3390/ijerph22060955

**Published:** 2025-06-18

**Authors:** Silvia Morales-Chainé, Gonzalo Bacigalupe, Rebeca Robles-García, Alma Luisa López-Fuentes, Violeta Félix-Romero

**Affiliations:** 1Facultad de Psicología, Universidad Nacional Autónoma de México, Mexico City 04510, Mexico; alma.luisa18@gmail.com (A.L.L.-F.); violeta.flix@gmail.com (V.F.-R.); 2Department of Counseling and School Psychology, College of Education and Human Development, University of Massachusetts Boston, Boston, MA 02125, USA; gonzalo.bacigalupe@umb.edu; 3Centro de Investigación en Salud Mental Global INPRFM-UNAM, Instituto Nacional de Psiquiatría Ramón de la Fuente Muñiz, Mexico City 14370, Mexico; reberobles@inprf.gob.mx

**Keywords:** perpetrating-intimate-violence, self-harm-suicide thoughts-behaviors, mental health, harmful-alcohol-use, cohort study

## Abstract

**Background** The COVID-19 epidemic had a deleterious impact on mental health and substance abuse and led to an increase in several forms of violence, including self-harm and interpersonal violence among youth from low- and middle-income countries. Nevertheless, the relationship between the variables and their directionality has not been recognized. This study describes the relationship directionality between these variables among 18- to 20-year-old Mexican youths during the COVID-19 pandemic. **Methods** The longitudinal cohort study comprises an evolving group of young Mexican adults: 1390 participants aged 18 in 2021, 654 aged 19 in 2022, and 442 aged 20 in 2023. Proportions by sex—50% were matched in every cohort, and the evolution–age sample accomplishment accounted for 47% in 2022 and 32% in 2023. **Results** According to a structural equation model, which fit the data from 195 iterations with 246 parameters (*X*^2^[2722] = 8327.33, *p* < 0.001), yielding a CFI of 0.946, a TLI of 0.943, and an RMSEA of 0.029 [0.028–0.029]), perpetrating intimate violence, preceded by suffering intimate violence, combined with suffering anxiety symptoms, was associated with self-harm–suicide thoughts and behaviors (ShSTB), marked distress, dysfunction, and somatization symptoms. The relationship was stronger in women and 20-year-old Mexicans. In men, this pathway was exclusively associated with ShSTB. Suffering from intimate violence has been associated with depression, anxiety, and PTSD symptoms, as well as harmful alcohol use. **Conclusions** During an epidemic, prevention programs should be designed to warn about self-harm–suicide thoughts and behaviors, not only to ensure the safety of the victims of intimate personal-violence but also to prevent the suicidal behavior of perpetrators.

## 1. Introduction

The COVID-19 pandemic directed attention to self-harm and suicidal thoughts and behaviors (ShSTB), in addition to the increase in perpetrating–suffering intimate-interpersonal violence, mental health symptoms, and harmful alcohol use (HAU) in youth from low and middle-income countries (LMICs). Previous data suggest a relationship between perpetrating–suffering intimate violence and ShSTB, with alcohol use and mental health symptoms as associated factors. Describing trends and relationship directionality between violence, ShSTB, mental health symptomatology, and HAU is needed to contribute to the development of effective self-harm–suicide prevention programs in young people in LMICs suffering or perpetrating violence.

The mental health Gap Action Programme [1] has stated that suicide is the act of deliberately killing oneself, and that self-harm is intentional self-inflicted poisoning or injury, which may or may not have a fatal intent or outcome. In such context, suicide is the fourth leading cause of death among 15–29-year-olds, with 77% of global suicides occurring in LMICs [2]. In Mexico, self-harm accounted for 5.3 (95% UI 4.28–6.46) deaths per 100,000 population in 2019, showing an upward trend since 2000 [3], with 2.2 (95% UI 1.73–2.70) and 8.7 deaths per 100,000 population (95% UI 7.06–10.60) for women and men, respectively. These disease burden levels place Mexico in the 3.40 to 60 quintiles across all countries. In a previous study of 18,449 Mexicans during the COVID-19 pandemic, we reported that 33.30% of our participants presented at least one ShSTB symptom, 4.20% had had previous self-harm thoughts and behaviors (PShTB), while 38.30% reported marked distress, dysfunction, and somatization (MDDS) [4].

The World Health Organization (WHO) [5] defined violence as occurring among individuals who may or may not know each other, including a range of acts from bullying and physical fighting to more severe sexual and physical assault. Youths aged 10–29 have experienced at least one type of assault [5]. The United Nations Office on Drugs and Crime Research—Data Portal—Violent and Sexual Crime (UNODC [6]) indicated a tendency for sexual violence based on the unique violent offense data available for Mexico, with a rate of 2.47 per 100,000 population in 2015, 3.46 in 2016, 2.89 in 2017, 2.82 in 2018, 1.96 in 2019, 3.71 in 2020, 6.00 in 2021, and 7.22 in 2022. In a recent study of 7420 Mexicans aged 18–24, we found that 23.48% of young participants perpetrated interpersonal violence and 15.38% perpetrated intimate violence, while 25.26% had experienced intimate violence in 2023 [7].

Regarding the American Psychiatric Association [8], Major Depressive Disorder is when sadness, interest loss, guilt or worthlessness, energy loss, concentration loss, appetite change, psychomotor agitation or retardation, sleep change, and suicidal thoughts occur, lasting for at least two weeks [8]. Generalized Anxiety Disorder is defined as symptomatology referring to excessive worry most days, and three out of six symptoms, such as restlessness, fatigue, decreased concentration, irritability, tension, or insomnia, that interfere with work or social functioning, all for at least six months [8]. Regarding Posttraumatic Stress Disorder, APA [8] has identified four groups of symptoms after an exposure to direct or witnessed threats, injuries, or violence: reexperiencing, avoidance of stimuli, negative alteration in cognition and mood, and marked alterations in arousal and reactivity, all related to the traumatic event. In this context, it has been described that the COVID-19 pandemic headed a 27.6% increase in depression cases and a 25.6% rise in anxiety in 2021 [9]. The Pan American Health Organization [10] also reported a 6.5% rise in mental health issues (8.9% in women and 3.2% in men) in Mexico between 2000 and 2019. The rate was 1716.4 YLD per 100,000 Mexicans in 2019 (1975.7 for women and 1438.2 for men). Our preliminary study during the pandemic indicated a prevalence of depression, anxiety, and PTSD symptoms among Mexican youth of 44.46%, 47.90%, and 29.47%, respectively [7].

Lastly, HAU is a pattern of psychoactive substance use that damages health. This damage may be physical or mental, and it is associated with social consequences, e.g., family or work problems [1]. Alcohol consumption is responsible for 2.6 million deaths each year globally. It is the leading factor for premature mortality and disability among those aged 20 to 39, accounting for 13% of the deaths in this age group [11]. In Mexico, alcohol use went up from 28% in 2011 to 33.6% in 2016 [National Report on Alcohol and Tobacco—Ramón de la Fuente Muñiz National Institute of Psychiatry [INPRFM], [12]). Our study showed that 20.20% of Mexican youth reported HAU during the pandemic [7].

The frequency of ShSTB, MDDS, perpetrating–suffering intimate–interpersonal violence, depression, anxiety, PTSD, and HAU between our Mexican population highlights the need to describe the relationship directionality between them, as well as the role of sex, for improving cost-effective preventive programs and mental health interventions. Clare et al. [13] have suggested that research should consider the common social determinants of violence and ShSTB, including early exposure to substance use and mental health issues. Perpetrating intimate violence, after suffering it, may explain the presence of the ShSTB. Comorbidity of suicide, self-harm, violence, mental health, and HAU may have an order of co-occurrence in life.

Perpetrating intimate violence and having ShSTB have common risky social determinants, including early exposure to adversity (such as witnessing or being a victim of intimate violence), substance use, and mental health problems [13]. Rooney et al. [14] even indicated that the experience of suffering and perpetrating violence may provide youth with the capability (such as fearlessness and physical means) to engage in suicidal behavior. They have particularly suggested that perpetration of violence may increase after being a victim of violence, and that such trajectory may increase the probability of suffering from ShSTB occurrence.

There is evidence to support the association between various psychosocial conditions (such as depression, hopelessness, social isolation, and adverse life events) and suicidal thoughts. Yu and Zhu [15] suggested that being the victim of intimate partner violence is associated with ShSTB. Aloyce et al. [16] also found that depression is a common health problem strongly associated with intimate partner violence perpetration in young men aged between 18 and 24 years. We previously explored the relationship between depression, anxiety, and related MDDS with ShSTB by sex in Mexican adults during the COVID-19 pandemic [4]. Our findings suggest that depression leads to ShSTB in three possible ways: through PShTB, PShTB affecting MDDS, and generalized anxiety affecting MDDS. Predisposing factors for ShSTB would appear to include depression, anxiety, and hopelessness. Nevertheless, Benatov et al. [17] reported that traditional bullying perpetration was prospectively associated with suicide attempts, whereas being the victim of bullying was cross-sectionally associated with suicide ideation and attempts.

Exposure to interpersonal violence is associated with suicide attempts and suicide among youth and young adults [18]. Among adults, those who perpetrate violence are more likely to report suicide attempts [19]. Research also examining non-suicidal and suicidal self-injury risk among victims of bullying (in other words, those who bully others and are bullied themselves) suggests that they are at greater risk for suicide than those who are only victims or perpetrators [20]. Wolford-Clevenger et al. [21] also found a high prevalence of non-fatal suicidal ideation among perpetrators of violence.

Kafka et al. [22] evaluated the nature of intimate violence prior to suicide, by sex, between 2010 and 2017 in North Carolina. They noted whether the person who died by suicide was described as a perpetrator of violence, a victim, or both (bilateral violence), recording the type of abuse, whether physical or emotional, verbal threats to harm/kill, illegal trespassing, sexual assault, or stalking, and abusive suicide-related threats in the two weeks prior to their death. They described 439 suicides in which 72.67% of male decedents had perpetrated non-fatal intimate violence compared to 51.22% of female decedents, who also reported perpetrating intimate violence. Findings suggested that intimate personal violence may be a precipitating factor in 6.1% of suicides in North Carolina. Over 80% of decedents had engaged in non-fatal intimate personal violence prior to their suicide, while intimate violence was more common for youth involved in the criminal legal system, who revealed suicidal intent and used a firearm or alcohol. A systematic review by Sesar et al. [23] identified nine studies examining the association between perpetrating intimate violence and non-fatal suicidal behaviors in community and clinical samples, concluding that suicide is an impulsive response to acute strain or conflict in an already abusive relationship.

Rooney et al. [14] examined the association between suffering, perpetrating, and attempting suicide in a national sample of American youth reporting suicide ideation in the past year, suggesting that those who both experience and perpetrate violence are at greater risk for engaging in suicide behaviors, with a cumulative effect. The authors worked with 821 participants who indicated that they had seriously thought about committing suicide in the past year—63.1% female, 64.7% white, and 12.1% hispanic. They used stepwise hierarchical regressions to examine the association between victimization and perpetrating violence and attempting suicide, reporting that 52.7% suffered some form of aggression, 30.8% perpetrated some form of violence, and 28% attempted suicide in the past year. In general, the authors concluded that victimization and the interaction between victimization and perpetrating violence were associated with the frequency of attempting suicide, over demographics, depression, and other non-violent externalizing behaviors.

While both being victim of and perpetrating violence are associated with suicidal behaviors generally, there is a need to research suicidal thoughts and behaviors among those who are both suffering and perpetrating violence [13]. Further research needs to confirm how perpetrating intimate violence is associated with single suicides unrelated to other violent deaths [17]. Additional research throughout a prospective longitudinal cohort study may help to describe the relationship between victimization and perpetration of violence and ShSTB over time. But mainly, we want to know: does perpetrating intimate violence, after suffering it, relate to the presence of the ShSTB, mental health problems, and HAU in life?

Prevalence of ShSTB, intimate violence, mental health symptoms, and HAU was high in Mexico during the pandemic. Previous independent studies also suggest an association between ShSTB and suffering–perpetrating intimate–interpersonal violence, mental health conditions, and HAU in Mexico [4,7]. A cohort longitudinal study will help to describe the relationship directionality between perpetrating and suffering intimate violence, ShSTB, mental health symptoms, and HAU among 18- to 20-year-old Mexican youths. In previous findings (e.g., [4,7,14] or [23]), we have proved directionality associations by screening for violence to ShSTB, mental health problems, and HAU. Thus, our hypotheses state that perpetrating intimate violence and suffering intimate violence are associated with ShSTB (Ha1 and Ha2), as well as depressive, anxious, and PTSD symptoms (Ha3, Ha4, and Ha5), and HAU (Ha6) in one evolving age group.

## 2. Methods

### 2.1. Participants and Procedure

We based our research on a longitudinal evolving group cohort study. Therefore, we analyze data from 2486 young Mexicans, comprising 1390 aged 18 in 2021 (55.91%), 654 aged 19 in 2022 (26.31%), and 442 aged 20 in 2023 (17.78%). As for proportions by sex, 50% were matched in every cohort. Thus, the evolution–age sample achievement accounted for 47% in 2022 and 32% in 2023 (see Table 1).

Participants were invited to enroll voluntarily in a web-based application called *My Health is also Mental* (https://www.misalud.unam.mx (accessed on 31 December 2023), [4]), through the Health Ministry and the University Call for applications on their websites (announced by press conferences on the radio, television, and internet). Participants entered if they were interested in screening their mental health or seeking psychological care during the COVID-19 pandemic. Therefore, once they entered the application, they had to use their email addresses to register and complete an evaluation of their mental health status and receive free feedback and treatment if necessary.

We used the criteria for internet E-surveys, such as data protection, development, testing, contact mode, advertising the survey, compulsory/voluntary participation, completion rate, cookies used, IP check, log file analysis, registration, and atypical timestamp considerations [24]. Since the survey platform automatically eliminates respondents who fail to complete the survey, we can only obtain complete response rates. It means that we have a zero rate of respondents who did not complete the survey. Consequently, we are not able to describe Mexicans not interested in participating in the strategy.

The Ethics Committee of the Psychology Faculty of the National Autonomous University of Mexico (UNAM) approved the project (code FPSI/422/CEIP/157/2020. All participants signed the informed consent about findings being used for epidemiological research and the opportunity to refuse complying with the data requests and dropping out at any point in the study. The study did not offer any incentive; however, it provided psychoeducational material, and written feedback (e.g., with infographics about relaxation techniques). Feedback contained information about how to get remote psychological care from public health services, if needed. Lastly, we asymmetrically transformed personal information from the database with numbers and letters to protect them while holding it in the official university domain. Only authorized researchers could access information using security locks that protect data.

### 2.2. Instruments

The survey [4] includes a sociodemographic section and seven psychological tests (see Appendix A). The self-harm/suicide survey is a twenty-one question yes/no dichotomic response item scale classified on three blocks referring to symptoms now, in the past month, and in the past year from the mhGAP algorithm [1,4]: the ShSTB warning signs, the PShTB, and the MDDS scales. Subjects are considered at risk when they have at least one symptom in each scale. The three-factor model obtained an $X^{2}$(186) = 19,643.19, *p* < 0.001; an *RMSEA* = 0.075 (0.074–0.076), a *CFI* = 0.984, a *TLI* = 0.982, and an α = 0.94.

The Life Events Checklist 5th edition (LEC-5; [25,26]) screens suffering–perpetrating intimate–interpersonal violence with 11 yes/no dichotomic response items relating to the past six months (see Appendix A). The suffering intimate violence scales obtained an $X^{2}$(2) = 6.09, *p* < 0.048, an *RMSEA* = 0.017 (0.001–0.032), a *CFI* = 0.999, a *TLI* = 0.997, and an α = 0.76 (Morales-Chainé et al. [7]). The suffering interpersonal violence scales got an $X^{2}$(13) = 343.57, *p* < 0.001, an *RMSEA* = 0.059 (0.053–0.064), a *CFI* = 0.949, a *TLI* = 0.917, and an α = 0.76. The perpetrating intimate model identified exact parameters, with an α = 0.68. The perpetrating interpersonal violence scales got an $X^{2}$(2) = 94.634, *p* < 0.001, an *RMSEA* = 0.079 (0.066–0.093), a *CFI* = 0.952, a *TLI* = 0.855, and an α = 0.68. Participants were asked to choose the violent event that bothered them most at the time, and to answer the questions in part B of the Posttraumatic Check List (PCL-5).

The PCL-5 included 20 five-option-response items [27] about stressful symptoms in the past month (see Appendix A). We used the four-factor DMS-5-TR [8,26] structure [28]: reexperiencing, avoidance, negative alterations in cognition and mood (NACM), and hyperarousal. The four-factor structure got an $X^{2}$(162) = 4535.593, *p* < 0.001, a *CFI* = 0.916, a *TLI* = 0.901, an *RMSEA* = 0.076 (0.074–0.078), an *SRMR* = 0.044, and an α = 0.93 [7]. PTSD is considered when the 2-response option or more is selected for at least one of the B-items, one of the C-items, two of the D-items, and two of the E-items, plus considering symptoms present for over a month.

The Major Depressive Episode (MDE) checklist is an 11-question five-option-response item scale covering the past twelve months from the DSM-5-TR [8] (see Appendix A). The scale obtained an $X^{2}$(42) = 1210.25, *p* < 0.001, a *CFI* = 0.954, a *TLI* = 0.939, an *RMSEA* = 0.072 (0.069–0.076), an *SRMR* = 0.034, and an α = 0.89 [7]. The at-risk score is met when part 1, part 2, part 3, and criterion A and B guidelines are reached. The criterion for Part 1 is met when items one and two are answered with options 1 or 2. The criterion for Part 2 is met when five or more items are answered with options 1 or 2 from items 2 to 10 and Part 1. The criterion for Part 3 is met when question 3 is answered with response options 1 or 2. Criterion A is met when part 1 and part 2 or 3 are completed. Criterion B is met when question 11 is answered with response options 1, 2, or 3. Lastly, an MDE is identified when criteria A and B are met [8].

The Generalized Anxiety (GA) [29] is a five-question eleven-response-options item scale about symptoms in the past two weeks (see Appendix A). The scale obtained an $X^{2}$(5) = 74.940, *p* < 0.001, a *CFI* = 0.998, a *TLI* = 0.995, an *RMSEA* = 0.043 (0.035–0.052), an *SRMR* = 0.005, and an α = 0.93 [7]. At risk GA resulted in a symptomatology average of 60%.

The WHO Alcohol, Smoking, and Substance Involvement Screening Test determines harmful use for ten groups of substances (e.g., HAU by screening use of alcoholic beverages [beer, wine, spirits], see Appendix A [30]). In the ASSIST, eight questions screen for HAU (e.g., failing to do what is expected because of the use of the drug in question); ASSIST also screens for injecting any drug (non-medical use only). The total ASSIST obtained an $X^{2}$[8] = 163,646, *p <* 0.001, an *RMSEA* = 0.051, confidence interval of 0.045–0.058, an *SRMR* = 0.018, a *CFI* = 0.986, and a *TLI* = 0.973 [7]. The Cronbach α varied from 0.71 for the hallucinogen dimension, 0.77 for alcohol, to 0.96 for opioids [7]. We have calculated the score of the HAU as instructed by the WHO [30] [on page 32 of the ASSIST manual], adding the answers to questions two to seven. In the study, we just considered the alcohol scale.

### 2.3. Data Analysis

First, we ran the Confirmatory Factor Analysis CFA for each scale (with the maximum likelihood [ML] and the diagonally weighted least squares [DWLS] procedures [31,32]. We evaluated the overall fit of the models with the chi-square goodness of fit test: the Comparative Fit Index (CFI), the Tucker–Lewis Index (TLI), the Root Mean Square Error of Approximation (RMSEA), and the Standardized Root Mean Square Residual (SRMR) [31,32]. We did not consider the SRMR index for categorical data [30]. We calculated Cronbach’s alpha reliability for each scale.

Secondly, we obtained scores for each scale and classified subjects who met the self-harm/suicide (ShSTB, MDDS), violence (LEC-5), PTSD (PCL-5), depression (MDE), anxiety (GA), and HAU (from the ASSIST) criteria for risk. We compared the distribution of participants regarding such criteria by sex and cohort year of the pandemic, performing chi-squared tests and considering *p*-values under 0.05 on distributions.

At the end, we evaluated the association directionality between suffering and perpetrating intimate and interpersonal violence, ShSTB, MDDS, mental health symptoms and HAU with the Structural Equation Modeling [SEM], with a mixture of continuous and categorical variables [31,32], based on the comparison with the chi-square curve. The chi-square test proves the hypothesized models fit the data, based on the good fitting data indexes (e.g., TLI), to be confident about the latent variable association directionality [32]. We also got models by sex and cohort year of the pandemic. For the analysis, we used the Lavaan statistical package 0.6–11, from the RSTUDIO^®^ statistical software 2022.02.0 [33] and the SPSS Statistics 25 [34].

## 3. Results

### 3.1. Confirmatory Factorial Analyses and Cronbach’s Alpha

Appendix B shows the psychometric properties of all scales in the present study, as a replication of previous studies [4,7]. Thus, Appendix B shows the CFAs of the ShSTB, MDDS, LEC-5, MDE, GA, PCL-5, and ASSIST scales. Data fitting showed *CFIs* and *TLIs* > 0.90, *RMSEAs* < 0.08, and *SRMRs* < 0.06—the latter for continuous data for all scales. The reliability of the scales ranged from 0.38 for the Perpetrating Interpersonal Violence scale to 0.97 for the NACM from the PCL-5 and the MDE.

### 3.2. Self-Harm–Suicide, Violence, Depression, Anxiety, PTSD, and HAU for the Total Sample by Sex and Cohort Years

The distribution of participants at risk for ShSTB, MDDS, violence, depression, anxiety, PTSD, and HAU in the total sample by sex and cohort year are shown in Table 2. In the overall sample and according to the cutoff score in the corresponding scales, 38.10% of participants were at risk for ShSTB and 40% for MDDS; 37.70% of youths were at risk for suffering intimate violence, 58.70% for suffering interpersonal violence, 19.10% for perpetrating intimate violence, and 32.50% for perpetrating interpersonal violence. 71.00% of participants were at risk for depression, and 47.70% for anxiety, 43.10% were reexperiencing violence symptoms, 36.10% suffered from avoidance, 47.40% from NACM, and 46.90% from hyperarousal, while 47.10% of participants met the PTSD criterion. Finally, 18.20% of participants were at risk for HAU.

Moreover, women at risk for ShSTB, MDDS, suffering–perpetrating intimate–interpersonal violence, and mental health symptoms were significantly higher than men (*p* < 0.001). The participants at risk for alcohol use were similar by sex. There were significant differences between the proportions of participants at risk in the three-year-cohort group for almost all variables except depression (*p* < 0.001). The trends would appear to have increased by the last cohort year of the pandemic: 20-year-olds in 2023. Depression symptoms remained stable across the three cohort years.

### 3.3. Structural Equation Modeling

Figure 1 shows the overall model. For the three years of the pandemic, the model includes the following paths: from perpetrating intimate violence (PIV) to ShSTB (β_PIV_, β_ShSTB_ = 0.137) and MDDS (β_PIV_, β_MDDS_ = 0.084), from anxiety to ShSTB (β_GA_, β_ShSTB_ = 0.591) and MDDS (β_GA_, β_MDDS_ = 0.634), and from suffering intimate violence (SIV) to reexperiencing, avoidance, negative alterations in cognitions and mood (NACM), hyperarousal (β_SIV_, β_Rex_ = 0.823, β_SIV_, β_Avo_ = 0.830, β_SIV_, β_NACM_ = 0.832, β_SIV_, β_Hyp_ = 0.829, respectively), to depression and anxiety (β_SIV_, β_MDE_ = 0.180, and β_SIV_, β_GA_ = 0.698, respectively), to HAU (β_SIV_, β_Alc_ = 0.347), and perpetrating interpersonal and intimate violence (β_SIV_, β_PIntV_ = 0.702, β_SIV_, β_PIntPV_ = 0.665, respectively).

SIV was indirectly associated with ShSTB and MDDS through PIV, meaning that it is associated with ShSTB and MDDS: SIV–PIV–ShSTB and SIV–PIV–MDDS, given that GA is also related to ShSTB and MDDS. The model fit the data from 195 iterations with 246 parameters (*X*^2^[2722] = 8327.33, *p* < 0.001). It yielded a CFI of 0.946, a TLI of 0.943, and an RMSEA of 0.029 [0.028–0.029]), using a mixture of continuous and categorical observed variables from the total sample. Appendix A shows factor loadings for the observed variables for each scale of the SEM. In all cases, factor loadings were greater than 0.300.

Table 3 shows the general pattern replicated for women and 20-year-old participants in 2023. For women and the oldest cohort, SIV + PIV and anxiety were therefore associated with ShSTB and MDDS symptoms, and SIV was strongly related to PTSD symptoms, anxiety, depression, HAU, PIPV, and PIV. For men, SIV + PIV and anxiety were related to ShSTB, but only anxiety was associated with MDDS symptoms. For men, SIV was also associated with mental health symptoms, HAU, and perpetrating interpersonal–intimate violence. For the two younger cohorts, anxiety was the variable related to ShSTB and MDDS symptoms. At the same time, SIV was associated with mental health symptoms, HAU, and perpetrating interpersonal–intimate violence. All models obtained CFIs and TLIs over 0.949, RMSEAs, and upper confidence intervals under 0.027.

The results suggest the scale’s dimensionality, perpetrating–suffering interpersonal–intimate violence, ShSTB, MDDS, mental health symptomatology, and HAU, among the cohort years during the pandemic, were all moderated by sex. We have evidence that perpetrating intimate violence and anxiety were related to ShSTB and MDDS (Ha1 and Ha4) for women, and in the last cohort year, SIV indirectly by PIV was also associated with ShSTB and MDDS (Ha2) for women and in the last cohort year. PIV and, indirectly, SIV were exclusively related to ShSTB for men (Ha1–Ha2). SIV was associated with PTSD symptoms (Ha5), depression (Ha3), anxiety (Ha4), HAU (Ha6), and perpetrating interpersonal–intimate violence for all participants.

## 4. Discussion

The present study describes the relationship directionality between perpetrating and suffering intimate violence, ShSTB, mental health symptoms, and HAU among 18- to 20-year-old Mexican youths through a longitudinal cohort study during the COVID-19 pandemic. Findings suggest an upward trend in self-harm–suicide, violence, mental health symptoms, and HAU in young Mexicans in 2021, 2022, and 2023. They mainly indicate that women and 20-year-old Mexicans who perpetrated intimate violence after suffering intimate violence and anxiety symptoms experienced self-harm–suicide warning signs and MDDS. Suffering intimate violence is also associated with PTSD, depression, anxiety symptoms, HAU, and perpetrating violence. These patterns varied for men and younger cohorts. In particular, men who also reported perpetrating intimate violence after suffering intimate violence and suffering anxiety symptoms experienced self-harm–suicide warning signs.

The proportions of young people perpetrating intimate violence were slightly higher than what our previous longitudinal cohort study suggested in 2024. We have seen asymmetries by sex for mental health conditions as well as patterns; except for depression, which remained high and stable during the pandemic for the entire sample [2,5,9]. HAU is a crucial condition associated with self-harm–suicide and violence. Findings indicated an increasing pattern among Mexican youth, reporting harm due to the use of this substance, and similar rates between men and women. Incidence of alcohol use was similar to what INPRFM [11] and our previous study indicated [7].

According to our hypothesis, findings thus suggest an associative pattern where perpetrating intimate violence, preceded by suffering intimate violence as well as anxiety symptoms, were associated with ShSTB and MDDS symptoms. However, the relationship was stronger in women and 20-year-old Mexicans. Among young Mexican men, the association between perpetrating intimate violence preceded by suffering violence and anxiety was exclusively associated with ShSTB warning signs. In addition, suffering intimate violence has been related to mental health symptoms such as depression, anxiety, and PTSD symptoms, as well as HAU. Our findings contribute to what Aloyce et al. [16] already suggested about depression related to intimate partner violence in youth, and to what Yu and Zhu [15] also proposed about intimate partner violence victimization related to ShSTB. Our findings indicate a central role played by the perpetrated intimate violence; once participants suffered from intimate violence, ShSTB warning signs and MDDS arose in women, and only ShSTB in men.

The central findings of PIV-SIV-GA associated with ShSTB and MDDS agree with previous ones reported by Benatov et al. [17] who also reported that perpetration was associated with suicide attempts, and being the victim of violence was related with suicide ideation and attempts. Our secondary proved hypothesis, in which SIV is associated with mental health and alcohol use, is also similar to what Kafka et al. [22] suggested about intimate violence being common in younger participants who also use alcohol and to our previous findings about SIV associated with depression, anxiety, PTSD symptoms, and HAU [7].

As Rooney et al. [14] suggested, in the current study, it appears we examined the association between perpetrating–suffering violence and self-harm–suicide thoughts and behaviors among Mexicans at the time of the evaluation. Nevertheless, instead of stepwise hierarchical regression, we used SEM analysis to examine these associations. Our model, therefore, suggested that the perpetration of violence was associated with ShSTB for the young Mexican people in our sample who experienced victimization, also supporting conclusions about the interaction between violent victimization and perpetration associated with ShSTB, even after accounting for sex, gender, mental health symptoms, and alcohol use.

Rooney et al. [14] suggested that separate processes contribute to suicide ideation and suicide attempts. They proposed that ideation-to-action frameworks could pinpoint the role of the acquired capability for suicide in the process of engaging in suicide behavior. They describe how painful or provocative experiences drive increases in pain tolerance and lack of fear of death, which contribute to the acquired capability for suicide. As all these authors have suggested, future studies should consider that being a victim of violence may be associated with increased pain tolerance. In contrast, violence perpetration may be associated with a decreased fear of death and an increased willingness to engage in self-directed violent acts. Morales-Chainé, Palafox, et al. [4] suggested distinguishing suffering from marked distress, dysfunction, somatization, and anxious depression from conditions where only the participants suffer from depression to prevent suicidal thoughts and behaviors. Moreover, current findings seem to suggest that previous experiences of suffering could interact with perpetrating intimate violence to explain reported ShSTB in both men and women and MDDS in women and 20-year-old participants.

It seems that being a victimi of intimate violence is followed by a traumatic stress response (such as reexperiencing, intrusive cognition, avoidance, or numbing) and that it may serve to habituate adolescents to pain, increasing their acquired capability and accounting for the association between victimization and suicide attempts [35]. Dewar et al. [36] also reported that perpetrators used non-fatal suicidal behaviors as a coping strategy to deal with overwhelming negative emotions, particularly in response to relationship stressors. Participants in our study reported this type of mental health symptom was associated with suffering intimate violence. It is, therefore, essential to continue exploring how mental health symptoms interact with perpetrating–suffering intimate violence to increase this proposed capability due to increased pain tolerance or a reduced fear of death in future research.

Meanwhile, Rooney et al. [14] have already suggested that repeated experiences of perpetration may result, for example, in decreased fear of death in the face of physical danger—signaling an increase in the acquired capability for suicide. A history of violent perpetration and victimization could indicate that the young participants in this study have an increased capability for self-harm–suicide thoughts and behaviors. However, it is essential to determine whether young Mexicans displaying an increase in marked distress, dysfunction, and somatization symptoms have greater pain tolerance or a reduced fear of death due to their acquired capability for perpetrating intimate violence after suffering intimate violence. There is a need to confirm these conditions and their relationship with the ShSTB warning signs in clinical studies and with men, women, and 20-year-old participants.

The Three-Step Theory (3ST; [37]) posits that ShSTB develops in the presence of everyday pain (such as emotional pain) connected with lack of hope in life. The second step is the development of severe ideation, when the pain becomes higher than the sense of connectedness. At the end, the theory emphasizes the capability for suicide as a component of the intent to die. The capability for suicide is characterized by the habituation to the fear of death aroused by suicide ideation. The exposure to factors associated with suicide behavior may facilitate the development of suicide capability and help us identify effective mechanisms for prevention and intervention on suicide behavior matters.

We were able to measure ShSTB, MDDS, perpetrating–suffering intimate–interpersonal violence, depression, anxiety, PTSD symptoms, and HAU in young Mexicans during the pandemic. ShSTB and MDDS symptoms were conceptualized in the mhGAP [1]. Perpetrating–suffering intimate–interpersonal violence was also conceptualized regarding what Oram et al. [38], Alexander and Johnson [39], Kourti et al. [40], Scott-Storey et al. [41], Weathers et al. [25], and Morales-Chainé et al. [7] suggested. Depression, anxiety, PTSD symptoms, and HAU were evaluated as recommended by the American Psychiatric Association [8], Goldberg et al. [29], Blevins et al. [28], and WHO [30], in that order.

Study findings suggest the relationship directionalities between ShSTB, MDDS, violence, mental health, and alcohol use for a group of young Mexicans during the pandemic. We have seen increased reports of self-harm, suicide risks, MDDS, perpetrating–suffering intimate–interpersonal violence, anxiety, PTSD symptoms, and alcohol use in youth aged from 18 in 2021 to 20 in 2023 at the end of the pandemic. Almost one in two youths suffered from ShSTB or MDDS symptoms. Findings showed that one in five women had perpetrated intimate violence. One in two women suffered intrafamilial violence as compared to one in three men [7]. Furthermore, 65.20% of the women suffered interpersonal violence as contrasting to 52.10 of men. ShSTB and MDDS, and suffering intimate and interpersonal violence, increased by the end of the pandemic among young people in our study.

Intimate violence precedes a significant proportion of ShSTB and MDDS, pointing to clear opportunities for intervention. It is essential to identify abusive partner-threatening suicide planning not only to ensure the safety of the victim of intimate personal violence but also to prevent the perpetrator’s suicidal behavior from emerging as a result of these social conditions. Considering trauma-informed approaches for intervention programs integrating suicide prevention efforts could increase the effectiveness of these programs while encouraging people who are perpetrating intimate personal violence to seek treatment [7]. Developing screening and early intervention for intimate personal violence and suicidal thoughts and behaviors mediated by sex could help prevent socially tough outcomes. We may provide training on assessment and early treatment of self-harm and suicide to those professionals that work with clients who suffer received or perpetrated violence.

## 5. Conclusions

Our study describes the relationship directionality between perpetrating and suffering intimate violence, ShSTB, mental health symptoms, and HAU among 18- to 20-year-old Mexican youths through a longitudinal cohort study during the COVID-19 pandemic. Findings suggest upward patterns in ShSTB, MDDS, violence, mental health symptoms, and harmful use of alcohol in very young Mexicans in 2021, 2022, and 2023. Nearly one in two youths suffered from ShSTB or MDDS symptoms. Furthermore, one in five women have perpetrated intimate violence, while one in two women suffered intimate violence in contrast to one in three men. Self-harm–suicide, marked distress, dysfunction, and somatization symptoms, and suffering intimate and interpersonal violence, increased by the end of the pandemic among young Mexicans. We noted asymmetries by sex for mental health conditions and growing patterns, except for depression, which remained high and stable during the pandemic, for our sample. Findings imply an increasing pattern among Mexican youth to report harm due to the use of this substance, with similar rates between young Mexican men and women.

Our study suggests an associative pattern where perpetrating intimate violence, preceded by suffering intimate violence combined with suffering anxiety symptoms were associated with ShSTB and MDDS. Nevertheless, the relationship was stronger in women and 20-year-old Mexicans. Among the young men in our sample, the relationship between perpetrating intimate violence preceded by suffering violence and anxiety was exclusively associated with ShSTB. Moreover, suffering from intimate violence has been related to mental health symptoms such as depression, anxiety, and PTSD symptoms, as well as HAU. Future research could explore whether young Mexicans displaying increased MDDS symptoms have increased pain tolerance or a reduced fear of death regarding their acquired capability for perpetrating intimate violence after suffering intimate violence. Confirming these conditions and their relationship with ShSTB in clinical studies and with men, women, and 20-year-old participants is essential.

Forthcoming research may consider mechanisms explaining how perpetrating–suffering violence is related to the capability for suicide behaviors and how MDDS symptoms increase tolerance of pain or reduce the fear of death. It may be helpful to study mechanisms to prevent in ShSTB and perpetrating–suffering intimate violence in Mexico. We need to adopt evidence-based interventions for self-harm and suicide while addressing perpetration of intimate violence at the primary care level, reducing the intervention gap to specialized care services in Mexico.

Intimate violence leads to a self-harm/suicide risk, emphasizing opportunities for prevention. We need to screen participants who warn about ShSTB for the safety of the victims of intimate personal violence and to prevent the perpetrator’s suicidal behavior. Conclusions in Mexican youth point to the need to design cost-effective perpetrating intimate violence interventions to prevent self-harm–suicide warning signs, marked distress, dysfunction, and somatization, distinguishing between sex and age, as well as to eliminate intimate violence and diminish mental health problems and HAU, as a public health policy for future pandemics. We need to design cost-effective intimate violence interventions to prevent ShSTB, MDDS, depression, anxiety, and HAU, integrating them into public health policy.

### Limitations

This is a longitudinal cohort study based solely on youth clinical reports of ShSTB, MDDS, violence, mental health symptomatology, and HAU. It has limitations in regard to the survey platform automatically eliminating respondents who fail to complete the screening. Thus, we may consider that we only obtained response rates of clinical relevance, and it forces us to reflect on the bias entailed in working with young Mexican’s data who were really seeking psychological care. In this context, our conclusions’ extent exclusively to clinical samples highly motivated to receive psychoeducation or treatment and not from a representative sample of Mexican youths. To generalize our findings, we suggest collecting data from a randomized sample of young Mexicans who are not essentially seeking professional help. Nevertheless, our findings are an alert about how mental health professionals may be aware of violence, self-harm/suicide, mental health, and HAU relationship while screening several mental health risks in the primary care settings.

Another limitation is the one regarding the need to include a wide range of Mexicans while determining the policies and community programs that should be implemented in Mexico. Upcoming studies may consider obtaining sensitivity and specificity evaluation research on the validity of our scales. Confirming these psychometric assessment characteristics during the pandemic would help solve their possible limitations. This would prevent overestimation of symptoms and reports while replicating the models in future research.

## Figures and Tables

**Figure 1 ijerph-22-00955-f001:**
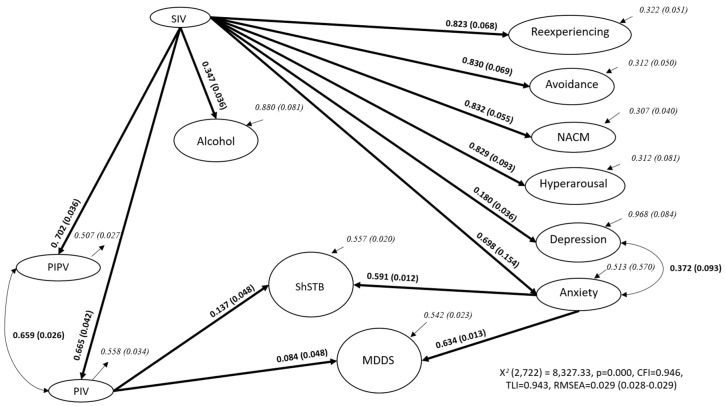
Variables from SEM, path coefficients, and residual variances for the whole sample. Adapted with permission from Morales-Chainé et al. [4]. *Note*. ShSTB—current Self-harm and Suicide Thoughts and Behaviors; MDDS—Marked Distress, Dysfunction, and Somatization; SIV—Suffering Intimate Violence; PIV—Perpetrating Intimate Violence; PIPV—Perpetrating Interpersonal Violence; NACM—Negative Alterations in Cognitions and Mood.

**Table 1 ijerph-22-00955-t001:** Proportions by Sex and Cohort Years.

Sex	18 Years 2021		19 Years 2022		20 Years 2023		Total	
	*n*	%	*n*	%	*n*	%	*n*	*%*
Men	695	50	327	50	221	50	1243	50
Women	695	50	327	50	221	50	1243	50
Total	1390	55.91	654	26.31	442	17.78	2486	100

**Table 2 ijerph-22-00955-t002:** Distribution of Participants at Risk for Self-harm–suicide, Violence, Depression, Anxiety, PTSD Symptoms, and Harmful Alcohol Use for the Total Sample by Sex and Cohort.

**ShSTB**												**MDDS**											
Total		Men		Women **		2021		2022		2023 **		Total		Men		Women **		2021		2022		2023 **	
n	%	n	%	n	%	n	%	n	%	n	%	n	%	n	%	n	%	n	%	n	%	n	%
946	38.10	362	29.10	584	47.00	500	36.00	223	34.10	223	50.50	995	40.00	392	31.50	603	48.50	542	39.00	227	34.70	226	51.10
**Suffering Intimate Violence**												**Suffering Interpersonal Violence**											
Total		Men		Women **		2021		2022		2023 **		Total		Men		Women **		2021		2022		2023 **	
n	%	n	%	n	%	n	%	n	%	n	%	n	%	n	%	n	%	n	%	n	%	n	%
938	37.70	356	28.60	582	46.80	475	34.20	245	37.50	218	49.30	1459	58.70	648	52.10	811	65.20	778	56.00	373	57.00	308	69.70
**Perpetrating Intimate Violence**												**Perpetrating Interpersonal Violence**											
Total		Men		Women **		2021		2022		2023 *		Total		Men		Women **		2021		2022		2023 **	
n	%	n	%	n	%	n	%	n	%	n	%	n	%	n	%	n	%	n	%	n	%	n	%
475	19.10	180	14.50	295	23.70	267	19.20	106	16.20	102	23.10	809	32.50	361	29.00	448	36.00	412	29.60	196	30.00	201	45.50
**Depression**												**Anxiety**											
Total		Men		Women **		2021		2022		2023		Total		Men		Women **		2021		2022		2023 **	
n	%	n	%	n	%	n	%	n	%	n	%	n	%	n	%	n	%	n	%	n	%	n	%
1765	71.00	853	68.60	912	73.40	976	70.20	467	71.40	322	72.90	1183	47.60	454	36.50	729	58.60	615	44.20	299	45.70	269	60.90
**Reexperiencing**												**Avoidance**											
Total		Men		Women **		2021		2022		2023 **		Total		Men		Women **		2021		2022		2023 **	
n	%	n	%	n	%	n	%	n	%	n	%	n	%	n	%	n	%	n	%	n	%	n	%
1072	43.10	423	34.00	649	52.20	523	37.60	285	43.60	264	59.70	897	36.10	342	27.50	555	44.70	437	31.40	231	35.30	229	51.80
**NACM**												**Hyperarousal**											
Total		Men		Women **		2021		2022		2023 **		Total		Men		Women **		2021		2022		2023 **	
n	%	n	%	n	%	n	%	n	%	n	%	n	%	n	%	n	%	n	%	n	%	n	%
1178	47.40	469	37.70	709	57.00	590	42.40	308	47.10	280	63.30	1166	46.90	466	37.50	700	56.30	578	41.60	311	47.60	277	62.70
**PTSD**												**Alcohol**											
Total		Men		Women **		2021		2022		2023 **		Total		Men		Women		2021		2022		2023 **	
n	%	n	%	n	%	n	%	n	%	n	%	n	%	n	%	n	%	n	%	n	%	n	%
619	47.10	223	40.90	396	51.40	293	44.30	152	43.20	174	57.60	453	18.20	236	19.00	217	17.50	186	13.40	142	21.70	125	28.30

*Note*. Adapted with permission from Morales-Chainé et al. [4]. n—number of participants; %—percentage of participants; ShSTB—current Self-harm and Suicide Thoughts and Behaviors; MDDS—Marked Distress, Dysfunction, and Somatization; NACM—Negative Alterations in Cognitions and Mood; PTSD—post-traumatic stress disorder; * Significant differences between groups < 0.05. ** Significant differences between groups < 0.001. At-risk group criteria were included in the instruments section. This means (1) a score over one for ShSTB, MDDS, suffering and perpetrating intimate and interpersonal violence is required to meet the criteria; (2) a score over 11 is required to meet the criteria for HAU; (3) criteria A and B for depression; (4) an average of over 60% for anxiety; (5) a two-response option or more is required for at least one of the B items, one of the C items, two of the D items, two of the E items, and bothersome symptoms for over a month for PTSD.

**Table 3 ijerph-22-00955-t003:** Variables from SEMs with their path coefficients (β) and standard errors (e) by sex and cohort year.

Predicted Variables	Men		Women		18-Year 2021		19-Year 2022		20-Year 2023	
	n = 1243		n = 1243		n = 1390		n = 654		n = 442	
	β=	e=	β=	e=	β=	e=	β=	e=	β=	e=
PIV										
ShSTB	0.254	0.147	0.332	0.103					0.269	0.07
MDDS			0.262	0.107					0.172	0.08
Anxiety										
ShSTB	0.489	0.019	0.410	0.014	0.681	0.021	0.596	0.028	0.460	0.020
MDDS	0.571	0.018	0.472	0.015	0.732	0.022	0.633	0.028	0.504	0.022
SIV										
Reexperiencing	0.837	0.099	0.901	0.109	0.848	0.082	0.881	0.150	0.834	0.191
Avoidance	0.843	0.104	0.902	0.110	0.856	0.083	0.881	0.155	0.834	0.208
NACM	0.837	0.088	0.917	0.085	0.859	0.061	0.886	0.120	0.854	0.190
Hyperarousal	0.842	0.129	0.907	0.152	0.849	0.122	0.893	0.194	0.851	0.250
Depression	0.177	0.069	0.153	0.049	0.207	0.051	0.146	0.083	0.155	0.081
Anxiety	0.709	0.240	0.582	0.183	0.741	0.214	0.729	0.295	0.564	0.323
Depression–Anxiety	0.389	0.162	0.281	0.099	0.382	0.134	0.414	0.195	0.354	0.158
Alcohol	0.403	0.063	0.376	0.054	0.360	0.047	0.347	0.036	0.283	0.091
PIPV	0.624	0.069	0.776	0.050	0.771	0.048	0.888	0.078	0.697	0.100
PIV	0.896	0.074	0.802	0.053	0.682	0.050	0.603	0.093	0.653	0.117
PIPV–PIV	0.141	0.096	0.855	0.031	0.854	0.036	0.851	0.064	0.586	0.063
Model										
*X* ^2^	4067.41		5274.5		4817.393		3552.16		3449.83	
df	2503		2722		2575		2575		2722	
*p*≤	0.001		0.001		0.001		0.001		0.001	
RMSEA	0.022		0.027		0.025		0.024		0.025	
Confidence Interval	0.021–0.024		0.026–0.029		0.024–0.026		0.022–0.026		0.022–0.027	
CFI	0.965		0.952		0.958		0.973		0.961	
TLI	0.964		0.949		0.956		0.971		0.959	

*Note*. n—number of participants; ShSTB—current Self-harm and Suicide Thoughts and Behaviors; MDDS—Marked Distress, Dysfunction, and Somatization; SIV—Suffering Intimate Violence; PIV—Perpetrating Intimate Violence; PIPV—Perpetrating Interpersonal Violence; NACM—Negative Alterations in Cognitions and Mood.

## Data Availability

The datasets generated during and/or analyzed during the current study are available from the corresponding author on reasonable request.

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
