# Peer review of "Perpetrating–Suffering Intimate Violence: Self-Harm–Suicide Thoughts and Behaviors, Mental Health, and Alcohol Use Among Mexican Youth During COVID-19"

_ijerph, 2025, doi:10.3390/ijerph22060955_

Round 1
Reviewer 1 Report
Comments and Suggestions for Authors
IJERPH REVIEW REPORT
I appreciate the opportunity to review the manuscript and congratulate the authors for the contribution they are making. With the aim of improving the document I make the following observations:
Regarding the objective, it is suggested to use the same wording at all ends of the manuscript. In some places they “synthesize” it, which generates confusion about it.
In section 1, Background, it is suggested that it be titled INTRODUCTION. Here it is recommended to present the definitions of the variables used and the respective dimensions that help contextualize the work in general. Furthermore, the complete term must be presented and then the abbreviations, as there is disorder in this. Later, only abbreviations can be used.
It is necessary to better work on the JUSTIFICATION of the work. Although there are two paragraphs of three lines each, it is necessary to expand and nourish the information.
Another aspect is dating. It can be seen that there is an excess of citations of references 11 and 13. Search for updated information to support the work.
Currently there is the DSM-5-TR. It is suggested to review it and update the information on what is used in DSM5.
42 references are presented, of which 10 correspond to scientific articles from the last 5 years. This represents 23% of the total references. However, of these references, 4 correspond to self-citations. This could constitute an ethical dilemma, since it is not advisable to include quotes from the same authors in the works. Therefore, there would only be 6 references from authors from the last 5 years. Therefore, it is recommended to carry out an exhaustive search in indexed databases for new information on the variables that support the work. It is perhaps the biggest observation that does not yet justify its publication. It is suggested that the percentage of referenced scientific articles reaches approximately 40%.
Author Response
|
Summary |
|
|
|
We want to thank you very much for taking the time to review our manuscript. Please find detailed responses below and the corresponding revisions/corrections highlighted/in track changes in the re-submitted files. |
||
|
Point-by-point response to Comments and Suggestions for Authors |
||
|
Comments 1: Regarding the objective, it is suggested to use the same wording at all ends of the manuscript. In some places they “synthesize” it, which generates confusion about it. |
||
|
Response 1: Thank you for pointing this out. We agree with this comment. Therefore, we have used the same wording at the discussion section – page 11, second paragraph, lines 447 to 450. “The present study describes the relationship directionality between perpetrating and suffering intimate violence, ShSTB, mental health symptoms, and HAU among 18- to20-year-old Mexican youths through a longitudinal cohort study during the COVID-19 pandemic.” |
||
|
We also have used the same wording at the conclusion section – page 14, Second paragraph, lines 572 to 575. “ Our study describes the relationship directionality between perpetrating and suffering intimate violence, ShSTB, mental health symptoms, and HAU among 18- to20-year-old Mexican youths through a longitudinal cohort study during the COVID-19 pandemic.” Comments 2: In section 1, Background, it is suggested that it be titled INTRODUCTION. Here it is recommended to present the definitions of the variables used and the respective dimensions that help contextualize the work in general. Furthermore, the complete term must be presented and then the abbreviations, as there is disorder in this. Later, only abbreviations can be used. |
||
|
Response 2: Agree. We have, accordingly, changed the title of section 1 to Introduction, in page 2, first paragraph, and line 38: “1. Introduction” We have also presented the suicide and self-harm definitions used to contextualize the work in general, in the introduction section, in page 2, second paragraph, and lines 48 to 50: “The mental health Gap Action Programme (1) has referred that suicide is the act of deliberately killing oneself, and that Self-harm is the intentional self-inflicted poisoning or injury, which may or may not have a fatal intent or outcome. In such context, …” We have presented the violence definition used to contextualize the work in general, in the introduction section, on page 2, third paragraph, and lines 60 to 62: “The World Health Organization [WHO, 4] defined violence to that occurring among individuals who may or may not know each other, including a range of acts from bullying and physical fighting to more severe sexual and physical assault.” We have presented the major depressive, generalized Anxiety, and posttraumatic stress disorder definitions used to contextualize the work in general, in the introduction section, in page 2, fourth paragraph, and lines 70 to 80: “Regarding the American Psychiatric Association (8), Major Depressive Disorder is the one where sadness, interest loss, guilt or worthlessness, energy loss, concentration loss, appetite change, psychomotor agitation or retardation, sleep change, and suicidal thoughts occur lasting for at least two weeks [8]. Generalized Anxiety Disorder is defined as symptomatology referred to excessive worry most days, and three out of six symptoms such as restlessness, fatigue, decreased concentration, irritability, tense, or insomnia, that interferes with work or social functioning, all for at least six months [8]. About Posttraumatic Stress Disorder, APA [8] has considered four groups of symptoms after an exposure to direct or witnessed threats, injuries, or violence: reexperiencing, avoidance of stimuli, negative alteration in cognitions and mood, and marked alterations in arousal and reactivity, all related to the traumatic event. In this context, it has been described that …” We have presented the harmful alcohol use definition used to contextualize the work in general, in the introduction section, on page 3, first paragraph, and lines 88 to 90: “Lastly, the HAU is a pattern of this psychoactive substance use that damages health. This damage may be physical or mental, and it is associated with social consequences, e.g. family or work problems [1].” We have presented the complete term about harmful alcohol use and then the abbreviation (HAU) later, along the paper. For example, in page 2, first paragraph, and line 41: “… and harmful alcohol use (HAU) in youth… » |
||
|
Comments 3: It is necessary to work better on the JUSTIFICATION of the work. Although there are two paragraphs of three lines each, it is necessary to expand and nourish the information. |
||
|
Response 3: Agree. We have, accordingly, expanded and nourished the justification. In the introduction section, page 3, second paragraph, on lines 101 to 104: “Perpetrating intimate violence, after suffering it, may explain the presence of the ShSTB. Comorbidity of suicide, self-harm, violence, mental health, and HAU may have an order of co-occurrence in life.” And in the third paragraph, on lines 110 to 112: “They have particularly suggested that perpetrating violence may increase after being a victim of violence, and that such trajectory may increase the probability of suffering from ShSTB occurrence.” |
||
|
Comments 4: Another aspect is dating. It can be seen that there is an excess of citations of references 11 and 13. Search for updated information to support the work. |
||
|
Response 4: Agree. We have, accordingly, updated information to support the work, reducing the excess of citations of reference of Rooney et al., and Kafka et al., and added new ones: Clare et al (2021), Yu, and Zhu (2023), Aloyce et al. (2024), and Benatov et al. (2022). First, it was added in the introduction section, on page 3, in the second paragraph, in lines 99 to 101. “Clare et al. [13] have suggested research should consider the common social determinants of violence and ShSTB, including early exposure to substance use and mental health issues.” On page 3, third paragraph, in lines 105 to 107: “Perpetrating intimate violence and having ShSTB have common risky social determinants, including early exposure to adversity (such as witnessing or being a victim of intimate violence), substance use, and mental health problems [13].” On page 3, in the fourth paragraph, in lines 115 to 118: “Yu and Zhu’s (15) suggested that victimization of intimate partner violence is associated with ShSTB. Aloyce et al. (16) also found that depression is a common health problem strongly associated with intimate partner violence perpetration in young men aged between 18 and 24 years.” On page 3, in the fourth paragraph, in lines 123 to 125: “Nevertheless, Benatov et al. [17] reported that traditional bullying perpetration was prospectively associated with suicide attempts, whereas bullying victimizing was cross-sectionally associated with suicide ideation and attempts.”
In the Discussion Section, in page 12, in the second paragraph, in lines 474 to 479:
“Our findings contribute to what Aloyce et al. [16] already suggested about depression related to intimate partner violence in youth, and to what Yu and Zhu [15] also proposed about intimate partner violence victimization related to ShSTB. Our findings indicate a central role played by the perpetrated intimate violence, once participants suffered from intimate violence over ShSTB warning signs and MDDS in women, and only over ShSTB in men.”
And in the Discussion Section, in page 12, in the third paragraph, in lines 481 to 483:
“Benatov et al. (17) who also reported that perpetration was associated with suicide attempts, and victimizing violence was related with suicide ideation and attempts.”
|
||
|
Comments 5: Currently there is the DSM-5-TR. It is suggested to review it and update the information on what is used in DSM5. |
||
|
Response 5: Agree. We have updated the information on what is used in the DSM-5-TR in the introduction section, on page 2, in the fourth paragraph, in lines 70 to 81: “Regarding the American Psychiatric Association (8), Major Depressive Disorder is the one where sadness, interest loss, guilt or worthlessness, energy loss, concentration loss, appetite change, psychomotor agitation or retardation, sleep change, and suicidal thoughts occur lasting for at least two weeks [8]. Generalized Anxiety Disorder is defined as symptomatology referred to excessive worry most days, and three out of six symptoms such as restlessness, fatigue, decreased concentration, irritability, tense, or insomnia, that interferes with work or social functioning, all for at least six months [8]. About Posttraumatic Stress Disorder, APA [8] has considered four groups of symptoms after an exposure to direct or witnessed threats, injuries, or violence: reexperiencing, avoidance of stimuli, negative alteration in cognitions and mood, and marked alterations in arousal and reactivity, all related to the traumatic event. In this context, it has been described that” in the instrument section, on page 6, in the second paragraph, in line 247: “We used the four-factor DMS-5-TR [8, 26] structure … » And, on page 6, in the third paragraph, in line 258: “…from the DSM-5-TR [8].” |
||
|
Comments 6: 42 references are presented, of which 10 correspond to scientific articles from the last 5 years. This represents 23% of the total references. However, of these references, 4 correspond to self-citations. This could constitute an ethical dilemma, since it is not advisable to include quotes from the same authors in the works. Therefore, there would only be 6 references from authors from the last 5 years. Therefore, it is recommended to carry out an exhaustive search in indexed databases for new information on the variables that support the work. It is perhaps the biggest observation that does not yet justify its publication. It is suggested that the percentage of referenced scientific articles reaches approximately 40%. |
||
|
Response 6: Agree. We have carried out the search in indexed databased, and actualized and added new references, and scientific articles. Changes may be observed in reference section, between pages 19 and 21, in lines 685 to 698: 15. “Yu Z, Zhu X. The association between victimization experiences and suicidality: The mediating roles of sleep and depression. Journal of Affective Discorder. 2023; 329, 243-250. https://doi.org/10.1016/j.jad.2023.02.093 16. Aloyce D, Stökkl H, Mosha N, Malibwa S, Hashim R, Ayieko P, Hapiga S, Mshana G. The association between depression, suicidal thoughts and intimate partner violence perpetration among young men in Mwanza, Tanzania. Journal of Family Violence. 2024; 1-13. https://doi.org/10.1007/s10896-024-00706-y 17. Benatov J, Brunstein KA, Chen-Gal S. Bullying perpetration and victimization associations to suicide behavior: A longitudinal study. European Child & Adolescent Psychiatry. 2022; 31, 1353-1360. https://doi.org/10.1007/s00787-021-01776-9 18. Berny LM, Tanner-Smith EE. Interpersonal violence and suicide risk: Ecamining buffering effects of school and Community connectedness. Children and Youth Services Review. 2024; 107405. https://doi.org/10.1016/j.childyouth.2023.107405 19. Mueller-Williams AC, Ilgen MA, Hicks BM. Characteristics of people who report firearm suicidal ideation in the USA. Inj Prev Epub ahead of print: [May 31,2025]. 2024. doi:10.1136/ip-2024-045341 20. Serafini G, Aguglia A, Amerio A, Canepa G, Adavastro G, Conigliaro C, Nebbia J, Franchi L, Flouri E, Amore M. The relationship between bullying victimization and perpetration and non-suicidal self-injury: A systematic review. Child Psychiatry & Human Development. 2023; 54:154-175. https://doi.org/10.1007/s10578-021-01231-5 “
And lines 748 and 749: 37. Klonsky ED, Pachkowski MC, Shahnaz A, May AM. The three-step theory of suicide: Description, evidence, and some useful points of clarification. Prev Med. 2021;152 (Pt 1):106549. https://doi.org/10.1016/j.ypmed.2021.106549
We have also eliminated two out of the four original self-citations, maintaining just information from two that are referring Mexican data. Changes may be observed in reference section, on page 19, lines 674 to 677: 4. “Morales-Chainé S, Palafox-Palafox G, Robles-García R, Arenas-Landgrave P, López-Montoya A, Félix-Romero A, Imaz-Gispert M. Pathways of depressive symptoms to self-harm and suicide warning signs during COVID-19 pandemic: The role of anxiety and related distress, dysfunction, and somatization. Journal of Affective Disorders. 2024; 35:476-484. https://doi.org/10.1016/j.jad.2023.12.077”
And in lines 681 to 683: “7. Morales-Chainé S, Bacigalupe G, Robles-García R, López-Montoya A, Félix-Romero V, Imaz-Gispert MA. Interpersonal and intimate violence in Mexican youth: Drug use, depression, anxiety, and stress during the COVID-19 Pandemic. International Journal of Environmental Research and Public Health. 2023; 20(15):6484. https://doi.org/10.3390/ijerph20156484” |
||

Reviewer 2 Report
Comments and Suggestions for Authors
The study is of interest, but the manuscript needs revisions.
There is no stated research question. As such, it is not clear what this study aims to accomplish. One sentence, on page 4 (lines 150-153) indicates that the study will report trends in perpetrating-suffering IPV and several risk factors. As such, the only guiding direction for this study claims IPV perpetration and victimization are associated with a variety of adverse conditions that have been readily studied. Perhaps there is something unique to the specific population under study, different from the current literature, but this is not clearly articulated. Further, it is not clear why the authors chose to combine both victimization and perpetration, as these are different phenomenon.
With this lack of clarification, I reviewed the methods only to ascertain a match between the intention of the study and methodology.
- How did the authors access the participants? And more specifically, how did the authors access the email address for participants?
- The design of the study is not adequately explained under the “design” section (2.1).
- How did the elimination of incomplete surveys impact the study? What percentage of respondents did not complete the survey (this is the actual response rate: completed/who took survey)? What are the descriptives of the respondents who did not complete the survey, compared to those who did complete? I wonder what degree of bias may be present due to failure to include all surveys?
- What are the psychometric properties of all scales in the present study? The authors report these properties within other studies but fail to report based on their sample.
- The authors failed to explain how IPV was measured, either victimization or perpetration. The measures are not listed in Appendix A. The authors need to explain why these 2 different phenomena were combined. There is no information on frequency.
- The authors claim to create an additive scale for the substance use scale (AOD). The measures for items 2-5 are different than 6-7, so these items cannot automatically be combined into an additive scale. Some transformation is necessary to make the measures compatible.
- There is no indication of how the authors determined directionality.
- There is repetition of sentences within sections 2.1 and 2.2.
- There is no stated hypothesis for scale testing, as indicated on page 7 (line 286).
- And finally, the statistical analyses used do not match the hypothesis: prevalence and directionality. What are the authors comparing by using chi-square? I am not sure this is the most appropriate statistical test: the authors do not provide a hypothesis indicating what they expect as an outcome, and there are not 2 variables being compared.
Overall, the study, as currently presented, does not support the hypothesis. Therefore, no other review was conducted. The authors need to offer clarification on the research question, review and revise the hypothesis, provide more information on the variables and sample, and revise the analyses to better suit the goal of the study.
Author Response
|
We want to thank you very much for taking the time to review our manuscript. Please find detailed responses below and the corresponding revisions/corrections highlighted/in track changes in the re-submitted files. |
|
Point-by-point response to Comments and Suggestions for Authors |
|
Comments 1: There is no research question stated. |
|
Response 1: Thank you for pointing this out. We agree with this comment. Therefore, we have stated a research question in the introduction section, on page 4, in the third paragraph, lines 168 to 170: “But mainly, we want to know: perpetrating intimate violence, after suffering it, may relate to the presence of the ShSTB, mental health problems, and HAU in life? ” |
|
Comments 2: As such, it is not clear what this study aims to accomplish. One sentence, on page 4 (lines 150-153) indicates that the study will report trends in perpetrating-suffering IPV and several risk factors. As such, the only guiding direction for this study claims IPV perpetration and victimization are associated with a variety of adverse conditions that have been readily studied. |
|
Response 2: Agree. We have, accordingly, précised the objective of the study, describing the relationships instead of reporting trends, in the introduction section, in page 4, in the third paragraph, in lines 166 to 168: “Additional research throughout a prospective-longitudinal cohort study may help to describe the relationship between victimizing and perpetrating violence and ShSTB over time.” And, on page 4, in the fourth paragraph, in lines 174 to 176: “A cohort-longitudinal study will help to describe the relationship directionality between perpetrating and suffering intimate violence, ShSTB, mental health symptoms, and HAU among 18- to 20-year-old Mexican youths.” |
|
Comments 3: Perhaps there is something unique to the specific population under study, different from the current literature, but this is not clearly articulated. |
|
Response 3: Agree. We have specified the population under study in the introduction section, on page 4, in the fourth paragraph, in line 176: “…among 18- to 20-year-old Mexican youths” |
|
Comments 4: Further, it is not clear why the authors chose to combine both victimization and perpetration, as these are different phenomenon. |
|
Response 4: Agree. We have, accordingly, clarified how the combinations may be explored because previous literature suggested in the introduction section, on page 3, in the second paragraph, between lines 104 and 104: “Perpetrating intimate violence, after suffering it, may explain the presence of the ShSTB. Comorbidity of suicide, self-harm, violence, mental health, and HAU may have an order of co-occurrence in life.” And, on page 3, in the third paragraph, between lines 110 and 112: “They have particularly suggested that perpetrating violence may increase after being a victim of violence, and that such trajectory may increase the probability of suffering from ShSTB occurrence.”
|
|
Comments 5: With this lack of clarification, I reviewed the methods only to ascertain a match between the intention of the study and methodology. |
|
Response 5: Agree. We have, accordingly and like repeatedly same wording the goal of the study in the abstract section, on page 1, in the first paragraph, between lines 17 and 18: “The study describes the relationship directionality between these variables among 18- to 20-year-old Mexican youths, during the COVID-19 pandemic.” In the introduction section, on page 4, in the fourth paragraph, between lines 174 and 179: “A cohort-longitudinal study will help to describe the relationship directionality between perpetrating and suffering intimate violence, ShSTB, mental health symptoms, and HAU among 18- to 20-year-old Mexican youths.” In the discussion section, on page 11, in the second paragraph, between lines 447 and 450: “The present study describes the relationship directionality between perpetrating and suffering intimate violence, ShSTB, mental health symptoms, and HAU among 18- to20-year-old Mexican youths through a longitudinal cohort study during the COVID-19 pandemic.” And, in the conclusion section, on page 14, in the second paragraph, between lines 572 and 574: “Our study describes the relationship directionality between perpetrating and suffering intimate violence, ShSTB, mental health symptoms, and HAU among 18- to20-year-old Mexican youths through a longitudinal cohort study during the COVID-19 pandemic.” |
|
Comments 6: How did the authors access the participants? And more specifically, how did the authors access the email address for participants? |
|
Response 6: Agree. We have included the information about how participants were invited, and how we get access to their email address, in the participants’ and design section, on page 5, second paragraph, between lines 190 and 197: “Participants were invited to enroll voluntarily in a web-based application called My Health is also Mental (https://www.misalud.unam.mx, [4]), through the Health Ministry and the University Call for applications on their Websites (announced by press conferences on the radio, television, and internet). Participants entered if they were interested in screening their mental health or seeking psychological care during the COVID-19 pandemic. Therefore, once they enter the application, they had to use their email addresses to register and complete an evaluation of their mental health status and receive free feedback and treatment if necessary.” |
|
Comments 7: The design of the study is not adequately explained under the “design” section (2.1). |
|
Response 7: Agree. We have rephrased the procedure of the study in the method section, on pages 4 and 5, last and first paragraph, respectively, between lines 183 and 188: “2.1. Participants and procedure We based our research on a longitudinal-evolving group cohort study. Therefore, we analyze data from 2,486 young Mexicans, comprising 1,390 aged 18 in 2021 (55.91%), 654 aged 19 in 2022 (26.31%), and 442 aged 20 in 2023 (17.78%). As for proportions by sex, 50% were matched in every cohort. Thus, the evolution-age sample achievement accounted for 47% in 2022 and 32% in 2023 (see Table 1). ” |
|
Comments 8: How did the elimination of incomplete surveys impact the study? |
|
Response 8: Agree. We have considered the impact of working with complete surveys in the limitation section, on page 15, in the second paragraph, between lines 616 and 627: “This is a longitudinal cohort study based solely on youth clinical reports of ShSTB, MDDS, violence, mental health symptomatology, and HAU. It has limitations in regard to the survey platform automatically eliminating respondents who fail to complete the screening. Thus, we may consider that we only obtained response rates of clinical matter, and it force to reflect about the bias of working with young Mexicans data who were really seeking psychological care. In this context our conclusions extent exclusively to clinical samples highly motivated to receive psychoeducation or treatment and not from a representative sample of Mexican youths. To generalize our findings, we suggest collecting data from a randomized sample of young Mexicans who are not essentially seeking professional help. Even though, our findings are an alert about how mental health professionals may be aware of violence, self-harm/suicide, mental health and HAU relationship while screening several mental health risks in the primary care settings.”
Comment 9: What percentage of respondents did not complete the survey (this is the actual response rate: completed/who took survey)? What are the descriptives of the respondents who did not complete the survey, compared to those who did complete? |
|
Response 9: Agree. We have, accordingly, include a statement about a cero rate of uncomplete surveys, in the participant section, on page 5, in the third paragraph, between lines 201 and 204: “Since the survey platform automatically eliminates respondents who fail to complete the survey, we only can obtain complete response rates. It means that we have a cero rate of respondents who did not complete the survey. Consequently, we are not able to describe Mexicans not interested in participating in the strategy.” |
|
Comments 10: I wonder what degree of bias may be present due to failure to include all surveys? |
|
Response 10: Agree. We have considered the bias of working with complete surveys in the limitation section, on page 15, in the second paragraph, between lines 616 and 627: “This is a longitudinal cohort study based solely on youth clinical reports of ShSTB, MDDS, violence, mental health symptomatology, and HAU. It has limitations in regard to the survey platform automatically eliminating respondents who fail to complete the screening. Thus, we may consider that we only obtained response rates of clinical matter, and it force to reflect about the bias of working with young Mexicans data who were really seeking psychological care. In this context our conclusions extent exclusively to clinical samples highly motivated to receive psychoeducation or treatment and not from a representative sample of Mexican youths. To generalize our findings, we suggest collecting data from a randomized sample of young Mexicans who are not essentially seeking professional help. Even though, our findings are an alert about how mental health professionals may be aware of violence, self-harm/suicide, mental health and HAU relationship while screening several mental health risks in the primary care settings.” |
|
Comments 11: What are the psychometric properties of all scales in the present study? The authors report these properties within other studies but fail to report based on their sample. |
|
Response 11: Agree. We have, accordingly, clarified that Appendix B shows the psychometric properties of the scales on the actual study, in the result section, on page 8, on the second paragraph, in lines 343 and 344: “Appendix B shows the psychometric properties of all scales in the present study, as replication on previous studies [4, 7]. Thus. …” |
|
Comments 12: The authors failed to explain how IPV was measured, either victimization or perpetration. The measures are not listed in Appendix A. The authors need to explain why these 2 different phenomena were combined. There is no information on frequency. |
|
Response 12: Agree. We have, accordingly, modified Appendix A, including Suffered Interpersonal Violence items, and using the whole name for Suffering Intimate Violence, Perpetrating Intimate Violence, and Perpetrating Interpersonal Violence, in the Appendix A section, on pages 15 and 16, and lines 653: And we have bold titles of table 2 to underline frequencies of violence types, in the result section, on pages 8 and 9, in line 361: |
|
Comments 13: The authors claim to create an additive scale for the substance use scale (AOD). The measures for items 2-5 are different than 6-7, so these items cannot automatically be combined into an additive scale. Some transformation is necessary to make the measures compatible. |
|
Response 13: We understand, thus We have, accordingly, clarified that we have adopted the WHO (30) ASSIST guide to calculate the harmful drug use scores, in the instrument section, on page 7, in the first paragraph, between lines 300 and 302: “We have scored the harmful use of each substance as the WHO [30] instructed to calculate it [on page 32 of the ASSIST manual]. In other words, we have calculated the HAU by adding the answers to questions two to seven.” |
|
Comments 14: There is no indication of how the authors determined directionality. |
|
Response 14: Agree. We have, accordingly, included statements in the introduction section, on page 3, in the second paragraph, between lines 102 and 104: “Perpetrating intimate violence, after suffering it, may explain the presence of the ShSTB. Comorbidity of suicide, self-harm, violence, mental health, and HAU may have an order of co-occurrence in life.” On page 3, in the third paragraph, between lines 110 and 112: “They have particularly suggested that perpetrating violence may increase after being a victim of violence, and that such trajectory may increase the probability of suffering from ShSTB occurrence.” And a brief declaration about directionality in the introduction section, on page 4, in the fourth paragraph, in lines 177 and 178: “Due to previous findings [e.g., 4, 7, 14, or 23], we have proved directionality associations by screening for violence to ShSTB – mental health problems – HAU. Thus, our…” |
|
Comments 15: There is repetition of sentences within sections 2.1 and 2.2. |
|
Response 15: Agree. We have corrected information about the survey in the instrument section, on page 5, in the fourth paragraph, and line 217: “The survey [4] was constituted by the sociodemographic section …” |
|
Comments 16: There is no stated hypothesis for scale testing, as indicated on page 7 (line 286). |
|
Response 16: We have bolded hypotheses statement, in the introduction section, on page 4, in the fourth paragraph, between lines 178 and 181: “Thus, our hypotheses state that perpetrating intimate violence and suffering intimate violence are associated with ShSTB (Ha1 and Ha2), as well as depressive, anxious, and PTSD symptoms (Ha3, Ha4, and Ha5), and HAU (Ha6) in one evolving age group.” |
|
Comments 17: And finally, the statistical analyses used do not match the hypothesis: prevalence and directionality. What are the authors comparing by using chi-square? I am not sure this is the most appropriate statistical test: the authors do not provide a hypothesis indicating what they expect as an outcome, and there are not 2 variables being compared. |
|
Response 17: To clarify, we have added a justification of the SEM, in the data analysis section, on pages 7 and 8, in the last and first paragraph, between lines 326 and 332: “To probe the models, we have used Structural Equation Modeling [SEM, 32] based on the comparison with the chi-square curve. The chi-square test provides information on the hypothesized models that fit the data. We have considered that every model approximates our reality. It means if every model is adjusted to data obtained, we may be confident about the association directionality probed between the variables. We again used the CFI, TLI, RMSEA and SRMR values as indicators of good data fit (32). Thus, …” |
|
Comments 18: Overall, the study, as currently presented, does not support the hypothesis. Therefore, no other review was conducted. The authors need to offer clarification on the research question, review and revise the hypothesis, provide more information on the variables and sample, and revise the analyses to better suit the goal of the study. |
|
Response 18: Agree. We have seriously considered the comments and added or corrected information about the research question, hypothesis, and analysis to better align with the study's goal. We appreciate the suggestions. |

Round 2
Reviewer 1 Report
Comments and Suggestions for Authors
Congratulations for the work done.